# CAN GANS RECOVER FAULTS IN ELECTRICAL MOTOR SENSORS?

**Sagar Verma,**[1,2] **Nicolas Henwood,**[2] **Marc Castella,**[3] **Jean-Christophe Pesquet,**[1]
**Al Kassem Jebai,**[2]
[1] Université Paris-Saclay, CentraleSupélec, Inria, Centre de Vision Numérique
[2] Schneider Toshiba Inverter Europe
[3] SAMOVAR, Télécom SudParis, Institut Polytechnique de Paris

## ABSTRACT

Electrical motors in industrial and emerging applications such as electrical automotive require high dynamic performance, robustness against parameter variation, and reliability. Recent advances in neural network-based estimators and fault detection techniques rely heavily on accurate sensor information. Due to the extreme operating conditions of electrical motors, there is always a chance of sensor failure which might lead to poor performance in downstream tasks using neural networks. This paper introduces the problem of identifying and recovering sensor faults using generative adversarial networks. We consider sensors monitoring various quantities like currents, voltages, speed, torque, temperature, and vibrations. We introduce fault model for these sensors to simulate training datasets. We use existing GAN based data imputation methods as baseline solutions.

## 1 INTRODUCTION

Electrical motors are widely used and constitute some of the most essential devices in heavy industries. These machines undergo various stresses in harsh conditions. A large number of adjustable speed drives in industry and emerging applications such as automotive and drones require high dynamic performance and robustness against parameter variations. As the functionalities of electrical motors have become increasingly complex, continuous monitoring using different sensors becomes necessary. Various monitoring and fault detection techniques exist that are based on system dynamics or are data-driven using machine learning and neural networks. Used sensors are rated for extreme operating conditions, but due to the nature of operations they may give out faulty data. Downstream monitoring and fault detection methods, especially neural networks, are very susceptible to faulty input and may deliver incorrect output.

For the proper work of modern electrical motors, mechanical and electrical variable sensors are necessary. Signals used in the internal control structure of the drive system (such as stator and/or rotor flux, electromagnetic torque, rotor speed) can be estimated by different simulators, observers, or neural networks (Verma et al., 2020b;a). The advanced control and fault detection mechanisms of electrical motor drives should be equipped with diagnostic features to prevent damages and sudden switch-offs of complex industrial installations. Thus the incipient fault detection has recently become one of the basic requirements for modern electrical motor drive systems (Jiang & Yu, 2012). For motor operation, current and voltage sensors are necessary for vector control algorithms (Orlowska-Kowalska & Dybkowski, 2010; Jiang, 2011). Rotor speed is also used in motor drive system recorded using mechanical or optical sensors (Fan & Zou, 2012). These are sensitive to the current drive and weather conditions (Gaeid et al., 2012). Traditional methods of monitoring and fault detection like (Benbouzid et al., 2007) rely on system mechanics, redundant sensors, and estimators to overcome sensor faults.

Neural networks (Che et al., 2018; Yoon et al., 2017; Cao et al., 2018) have been used to impute missing data. Generative Adversarial Networks (GAN) have been trained on time-series data which learn relationships between variables (Yoon et al., 2019). The same property of learning relationships between different variables in an input time series is then used to recover missing data (Yoon et al., 2018; Luo et al., 2018; 2019). We introduce the problem of detecting and recovering from erroneous and missing data. We present fault models for different types of sensors. We then use ex-

isting missing data imputation methods to solve the problem, discuss their limitations, and possible improvements.

## 2 PROBLEM FORMULATION

Existing neural network methods for missing data imputation use datasets which have either missing data or can be synthetically generated during training. To generate sensor faults, we first model the faults in different types of sensors using the process described in (Balaban et al., 2009). We consider following types of sensors: a) three current shunts measuring currents in three phase ($i_a, i_b, i_c$), b) voltmeters measuring three phase voltages ($u_a, u_b, u_c$), c) encoder measuring speed ($\omega_r$), d) torque meter measuring torques ($\tau$), e) temperature sensors ($\vartheta$), and f) accelerometers measuring vibrations ($\sigma$). Sensor faults can be classified into six categories as shown in (Zimmerman & Lyde, 1992; Iyengar & Prasad, 1995). These faults can be used to generate erroneous and missing values using nominal and abnormal statistics provided in the literature and sensor datasheets.

- **Bias:** A constant offset from the nominal sensor signal statistics given by $Y_f = X + \beta +$ noise, where $\beta$ is the constant offset value, $X$ is true value, $Y_f$ is the faulty value, and noise is a disturbance within a tolerance range.

- **Drift:** A time-varying offset from the nominal sensor statistics given by $Y_f = X + \delta(t) +$ noise, where $\delta(t)$ is the time-varying offset factor.

- **Scaling:** Magnitudes are scaled by a factor, where the form of the waveform itself does not change. This is given by $Y_f = \alpha(t) X +$ noise, where $0 < \alpha(t) < \infty$ is a scaling constant that may be time-varying.

- **Noise:** A random time series is observed, $Y_f =$ noise

- **Hard Fault:** The sensor output is stuck at a particular level expressed by $Y_f = C +$ noise, where $C$ is a constant. Hard fault can be due to loss of signal ($C = 0$) or stuck sensor where $C$ is some non-zero constant. Hard faults are usually treated as missing values.

- **Itermittents:** Deviations from normal readings appear and disappear several times from the sensor signal. The frequency of such signatures is generally random.

To represent errors and missing values in time-series signals, consider a $d$-dimensional multivariate time series $x$, observed at $t = (t_0, ..., t_{n-1})$, denoted by $x = (x_0, ..., x_{n-1}) \in \mathbb{R}^{d \times n}$, where $t$ is the observing timestamp, and $x_t = (x_t^j)_{1 \leq j \leq d} \in \mathbb{R}^d$ is the $t$-th observation. Let $m \in \mathbb{R}^{d \times n}$ is a mask matrix that takes values in $\{0, 1\}$. The values signal whether the components of $x$ exist or not, for example, if $x_t^j$ exists then $m_t^j = 1$, otherwise it is equal to 0. $e \in \mathbb{R}^{d \times n}$ is an error matrix that takes values in range $\{e_j^l, e_j^h\}$, where $e_j^l$ and $e_j^h$ are the lowest and highest error possible errors in time, for every $j \in \{0, \ldots, n-1\}$. The purpose of multivariate time series imputation is to impute the missing values and correct the erroneous values in $x$ as accurately as possible. Some imputation methods require time delta information. We define a matrix $\delta \in \mathbb{R}^{d \times n}$ that records the time lag between the current value and the last observed one. The components of $\delta$ are defined as follows:

$$\delta_{t_i}^j = \begin{cases} 0, & \text{if } i = 0 \\ t_i - t_{i-1}, & \text{if } i \neq 0 \text{ and } m_{t_{i-1}}^j (1 - e_{t_{i-1}}^j) = 1 \\ \delta_{t_{i-1}}^j + t_i - t_{i-1}, & \text{otherwise,} \end{cases} \tag{1}$$

The following example provides an intuitive explanation of a multivariate time series $x$ and its corresponding $m$ and $e$ variables, where "Na" designates a missing value and $\underline{x_i^j}$ is an erroneous value:

$$x = \begin{bmatrix} 1 & 2 & 3 & \text{Na} \\ \text{Na} & 2 & \underline{3.2} & 4 \\ 1 & \text{Na} & 3 & \underline{3.9} \end{bmatrix} \cdots , m = \begin{bmatrix} 1 & 1 & 1 & 0 \\ 0 & 1 & 1 & 1 \\ 1 & 0 & 1 & 1 \end{bmatrix} \cdots , \tag{2}$$

$$e = \begin{bmatrix} 0 & 0 & 0 & 0 \\ 0 & 0 & 0.2 & 0 \\ 0 & 0 & 0 & -0.1 \end{bmatrix} \cdots , \delta = \begin{bmatrix} 0 & 1 & 1 & 1 \\ 0 & 1 & 1 & 2 \\ 0 & 1 & 2 & 1 \end{bmatrix} \cdots , t = (0, 1, 2, 3, \ldots). \tag{3}$$

# 3 EXPERIMENTS AND RESULTS

## 3.1 DATASETS

**Induction Motor dataset** (Verma et al., 2020b) consists of 100 hours of simulated and 20 minutes of real world motor data collected from a 4kW induction motor. The dataset has been used in (Verma et al., 2020b;a; 2022) for designing neural speed-torque estimators which predicts speed ($\omega_r$) and estimated torque ($\tau_{em}$) from input currents ($i_d$, $i_q$) and voltages ($u_d$, $u_q$).

**PMSM Temperature dataset** (Kirchgässner et al., 2021) has been used for data-driven thermal modeling to remove or reduce the cost of placing thermal sensors deep inside moving parts of motors. It consists of different experiments where temperature of the stator and rotor were measured in real operating conditions. The following motor quantities are present in the dataset: currents ($i_d$, $i_q$), voltage ($u_d$, $u_q$), speed ($\omega_r$), torque ($\tau_{em}$). For the inference task, the following temperatures have been collected: permanent magnet ($\vartheta_{PM}$), stator yoke ($\vartheta_{SY}$), stator tooth ($\vartheta_{ST}$), stator winding ($\vartheta_{SW}$), ambient temperature outside of stator ($\vartheta_a$), and coolant temperature ($\vartheta_c$).

**Broken Bar Detection dataset** (Maciejewski et al., 2020) consists of electrical and mechanical signals recorded from 0.7457kW three-phase induction motor. The dataset consists of currents and voltages represented in $abc$ frame. We convert it to $dq$ frame using Clarke-Park transformation as explained in (O'Rourke et al., 2019). The dataset has 40 experiments collected at 60Hz. Mechanical signals were collected using five axial accelerometers ($\sigma$). These sensors capture vibration measurements in both drive end (DE) and non-drive end (NDE) sides of the motor, axially or radially, in the horizontal or vertical directions. For the electrical signals, the currents were measured by alternating current probes. The voltages were measured directly at the induction terminals using voltage points of the oscilloscope.

## 3.2 DATA IMPUTATION METHODS

**Mean**: We replace the missing values with mean value of the sensor calculated from the dataset.

**KNN** (Hudak et al., 2008): The missing values are replaced by using the $k$ nearest neighbor samples. We try $k = 3, 5, 7$ when experimenting on speed-torque dataset and choose $k = 5$ considering time and compute cost.

**MF** (Acar et al., 2010): Matrix Factorization (MF) method is used to factorise the incomplete matrix into low-rank matrices and fill the missing values. After several experimentation, we use 100 epochs and a learning rate of 0.001 for MF.

**MICE** (White et al., 2011): Multivariate Imputation by Chained Equations (MICE) fills the missing values by using iterative regression model. Max number of iterations is set at 100.

**GRUD** (Che et al., 2018): GRUD is a recurrent neural network that uses weighted combination of Gated Recurrent Units (GRU) output, last observation, and global mean to impute missing data.

**M-RNN** (Yoon et al., 2017): M-RNN is a bi-directional RNN which uses hidden states in both directions of RNN to impute values. M-RNN does not consider the correlations among different missing values.

**BRITS** (Cao et al., 2018): This method uses bi-directional recurrent network to impute time series. It implicitly updates missing information and can be used directly for downstream tasks.

**GAIN** (Yoon et al., 2018): GAIN is a GAN based imputation method that uses a hint vector that closely matches missing vectors distribution to impute missing values.

**2Stage GAN** (Luo et al., 2018): 2Stage GAN trains GAN in two stages to impute missing data.

**$E^2$-GAN** (Luo et al., 2019): $E^2$-GAN is an end-to-end version that overcomes the inefficiency of 2stage training by using a single stage.

GRUD, M-RNN, BRITS, GAIN, 2Stage GAN, and $E^2$-GAN use $\delta$ defined in Equation equation 1.

| Fault | Range | Median | Sensors and References |
|---|---|---|---|
| **Bias** | 1.2% to 60%
0.1% to 0.15% | 20%
0.12% | $\vartheta$ and $\omega$ (Qin & Li, 1999)
$\tau$ (Magtrol, 2021) |
| **Scaling** | 2.5 to 4.8
0.3 to 0.7 | 3.28
0.45 | $\sigma$ (Zimmerman & Lyde, 1992)
$T$ (Wang et al., 1998) |
| **Drift** | 6% to 75%
0.1% to 0.2% | 29%
0.17% | $T$ (Goebel & Yan, 2008)
$\vartheta$ (Qin & Li, 1999)
$\tau$ (Magtrol, 2021) |
| **Noise** | 2.5% to 250%
0.1% to 2%
1% to 6% | 20%
0.17%
2.4% | $T$, $\tau$ and $\omega$ (Lu & Hsu, 2002)
$\sigma$ (Zimmerman & Lyde, 1993)
$\tau$ (Magtrol, 2021)
$i$ and $u$ (Zhang et al., 2013) |
| **Intermittent (Hard Fault)** | 2 to 10 drops | 8 drops | all sensors (Goebel & Yan, 2008) |

Table 1: Faults leading to missing and erroneous values in different types of sensors.

## 3.3 TIME SERIES IMPUTATION

For all our experiments we use an Ubuntu 20.04 OS with RTX 3090. PyTorch is employed to implement all networks. For RNN and GAN based methods we use training hyperparameters reported in the source literature. We track the Root Mean Square Error (RMSE) and the Mean Absolute Error (MAE) during GAN training, and train the network till MAE metrics converge. We sample input data with a window of size 100 and stride equal to 1. We generate missing mask $m$ with probability 0.25 and error $e$ with probability 0.1 for a sensor using the fault model shown in Table 1. For all the imputation methods, since there is no way of incorporating erroneous data, we treat them as missing.

| Dataset | Non NN | | | | RNN | | | GAN | | |
|---|---|---|---|---|---|---|---|---|---|---|
| | **Mean** | **KNN** | **MF** | **MICE** | **GRUD** | **M-RNN** | **BRITS** | **GAIN** | **2-Stage** | $E^2$**-GAN** |
| **IM** | 0.632 | 0.392 | 0.483 | 0.400 | 0.486 | 0.436 | 0.396 | 0.318 | 0.373 | **0.352** |
| **PMSM** | 0.672 | 0.432 | 0.523 | 0.440 | 0.526 | 0.512 | 0.436 | 0.412 | 0.413 | **0.372** |
| **Broken Bar** | 0.757 | 0.517 | 0.608 | 0.525 | 0.611 | 0.598 | 0.521 | 0.509 | 0.498 | **0.477** |

Table 2: MSE results for different imputation methods.

Row 1-3 in Table 2 show MSE between true and imputed values for Induction, PMSM, and Broken Bar datasets, respectively. Standard imputation methods like mean and KNN (K=5) based imputation can be applied directly on the signals and provide some good starting baselines. RNN based approaches like GRUD, M-RNN and BRITS perform at least as good as MF and MICE. We can see that GAN based methods provide best imputation and corrected values. $E^2$-GAN outperforms all other methods owing to its GRU for Imputation units (GRUI) used in encoder and decoder part of generator and discriminator networks. GRUI unit uses $\delta$ to introduce a time decay vector to decrease the memory of GRU cell.

## 4 DISCUSSIONS

The main challenge associated with the proposed problem is the difficulty associated with the collection of dataset since it requires availability of faulty sensors. Operating a complex industrial machine with faulty sensors for making datasets is not economical and can be detrimental to the machine. Proposed fault models can be used to create large amount of synthetic data with sensor faults derived from small amount of real data. Several existing data imputation methods like Mean and KNN can be applied directly. MF and MICE perform similar to RNN based methods but they are not realizable in real time due to large delays. GAN based methods outperform all other methods and have delays that make them feasible for use in real-time.

The major limitation of all the methods is that they cannot identify sensor faults that are grouped as errors. We treat erroneous data as missing data in our experiments which is not correct as we do not know erroneous data apriori. We believe that our question around *"Can GANs recover sensor faults?"* has been answered and now there is an open problem of extending GANs to work for both erroneous and missing data.

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
