# OpenReview forum: "Can GANs Recover Faults in Electrical Motor Sensors?"
_ICLR.cc/2022/Workshop/DGM4HSD — ICLR 2022 DGM4HSD workshop Poster_

### Official Review · Reviewer_dydy · 2022-03-19
**Interesting work with broad set of baselines and datasets**

**Rating:** 9
**Confidence:** 4

**Review:**

The paper adapts a general problem of data imputation on the specific application field of electrical motors. Overall the paper is well written and the amount of baselines and datasets is good to evaluate the paper's contribution.

I provide the following feedback as weaker points of the paper with some suggestions:
- In the abstract and introduction it is mentioned "parameter variation", but I'm not sure it is clear what this means in this context.
- The related work section of the paper seems to be lacking a better view on the specific field explored in the paper, as the methods they present on data imputation are the generic ones. How did the field of electrical motors impute data before?
- One of the faults considered by the authors is "noise". Is this noise Gaussian or could it come from other distributions? Would the models behave differently in these cases?
- In section 3.3 it is not clear how the authors split the datasets, which I think can be of particular importance when talking about time series. Have the authors considered using cross validation on the test sets to have a measure of variability across datasets and models? I believe this measure of variability would bring stronger results analysis to the paper.
- Possible typo: Right after Table 2, "Row 1-4" should probably be "Rows 1-3"?
- Recently I came across this other work on data imputation: https://www.frontiersin.org/articles/10.3389/fgene.2021.624128/full. Here, the authors developed a significantly simplified model (when compared to GAIN) called PMI, which was able to achieve quite good results. Besides, they differentiate between in-place and induction imputations. I think these two points of the paper I suggest could be useful for the authors if they continue working on this topic.

---

### Official Review · Reviewer_RRF4 · 2022-03-23
**Using gans to impute missing sensor data.**

**Rating:** 5
**Confidence:** 4

**Review:**

This paper recovers sensor faults by leveraging the distribution modeling capabilities of generative models. Using custom faults models, it first generates a training dataset for the generative models. Next, it considers 10 different data imputation mechanisms and shows that the generative adversarial networks (GAN) based approach outperforms other methods.

The paper provides extensive details on the experimental setup, including fault mechanism (table-1) and baseline data imputation methods (section 3.2). However, most of these details can be moved to the appendix leaving more room to discuss the key contributions of the paper, i.e., the efficacy of GANs in the proposed task.

This is even more critical since the paper lacks further insights into why a GAN model outperforms other deep learning-based methods, such as RNN. In addition, to improve reproducibility, I encourage authors to report the hyperparameters used to train the generative model.

---

### Official Review · Reviewer_M84q · 2022-03-25

**Rating:** 7
**Confidence:** 4

**Review:**

The paper studies how to use GAN to simulate the missing data for electrical motors and sensor data (e.g., time series).
The authors incorporate some classical time series modeling methods (e.g., MICE) for data simulation.
Although some neural network-based generative baselines are missing (e.g., autoencoder), I still the GAN-aware modeling is interesting as an empirical study. I recommend accepting this paper.

Some future suggestions could be if that possible to use some pre-trained models from other data-sufficient domains (e.g., speech in Voice2Series [1]) for cross-domain transfer or using some pre-trained image models. The authors are encouraged to include the related references in their final version to enlarge the impacts of this work on the general ML community.

***

**References**
1. "Voice2series: Reprogramming acoustic models for time series classification." International Conference on Machine Learning (ICML) PMLR, 2021.

2. "Improved Input Reprogramming for GAN Conditioning." arXiv preprint arXiv:2201.02692 (2022).

3. "A Study of Low-Resource Speech Commands Recognition based on Adversarial Reprogramming." arXiv preprint arXiv:2110.03894 (2021).

---

### Decision · Program_Chairs · 2022-03-26

Accept (Poster)